# Distinguishing the sources of silica nanoparticles by dual isotopic fingerprinting and machine learning

Xuezhi Yang[1,2], Xian Liu[1], Aiqian Zhang[1,2], Dawei Lu[1], Gang Li[1], Qinghua Zhang[1], Qian Liu[1,2,3] & Guibin Jiang[1,2]

One of the key shortcomings in the field of nanotechnology risk assessment is the lack of techniques capable of source tracing of nanoparticles (NPs). Silica is the most-produced engineered nanomaterial and also widely present in the natural environment in diverse forms. Here we show that inherent isotopic fingerprints offer a feasible approach to distinguish the sources of silica nanoparticles ($SiO_2$ NPs). We find that engineered $SiO_2$ NPs have distinct Si–O two-dimensional (2D) isotopic fingerprints from naturally occurring $SiO_2$ NPs, due probably to the Si and O isotope fractionation and use of isotopically different materials during the manufacturing process of engineered $SiO_2$ NPs. A machine learning model is developed to classify the engineered and natural $SiO_2$ NPs with a discrimination accuracy of 93.3%. Furthermore, the Si–O isotopic fingerprints are even able to partly identify the synthetic methods and manufacturers of engineered $SiO_2$ NPs.

[1] State Key Laboratory of Environmental Chemistry and Ecotoxicology, Research Center for Eco-Environmental Sciences, Chinese Academy of Sciences, Beijing 100085, China. [2] College of Resources and Environment, University of Chinese Academy of Sciences, Beijing 100190, China. [3] Institute of Environment and Health, Jianghan University, Wuhan 430056, China. Correspondence and requests for materials should be addressed to Q.L. (email: qianliu@rcees.ac.cn) or to G.J. (email: gbjiang@rcees.ac.cn)

Nanoparticles (NPs) in the environment can occur naturally or originate from engineered nanomaterials released by human activities. Nowadays, the production and disposal amounts of engineered NPs are increasing rapidly, which raises significant health and safety concerns about the use of NPs[1,2]. Distinguishing the sources of NPs is of extreme importance for nano research, especially in areas of nanotechnology risk assessment, NP exposure monitoring, environmental fate studies, and nano-product analysis[3,4]. Although engineered NPs are usually produced in high purity, it is still one of the most challenging tasks in nanoanalytics to detect/distinguish them in complex natural media. In earlier works, the sources of NPs were normally identified by the means of morphology or chemical composition (e.g., multi-element analysis at single-particle levels that provided a basis for distinguishing between natural and engineered NPs)[4,5]. However, the results are sometimes speculative and clear distinguishing criteria are still lacking.

Isotope ratios have been widely used as powerful tracers and chronometers in geoscience, archeology, anthropology, and environmental science[6,7]. Stable isotopic fingerprints (or signatures) of elements in samples may contain valuable information on sources and processes which can reflect the history of the samples[8]. For nanotechnology, stable isotopic tracing was also expected to be a valuable tool[9]. The natural transformation or industrial synthetic processes of NPs may cause stable isotope fractionation[10], so it is hypothesized that NPs of different origins may possess different isotopic fingerprints. Unfortunately, up to now, the use of stable isotopic fingerprints in source tracing of NPs has not succeeded. Only few studies investigated the Zn or Ce isotopic compositions of ZnO or CeO2 NPs, but they did not find distinct difference from natural and anthropogenic materials and thereby concluded that the detection of NPs in natural samples by stable isotopic tracing was not feasible[11,12].

Silica nanoparticles (SiO2 NPs) are the most produced engineered nanomaterials (global production volume 185–1400 kilotons in 2014[13,14]) and extensively used in construction materials, microelectronics, food and pharmaceutical industries, and consumer products[15]. However, SiO2 NPs have also been shown to pose a significant risk to human health, e.g., inhalation of SiO2 NPs can lead to severe inflammation of the respiratory system and systemic autoimmune diseases[16–18]. On the other hand, natural silica is ubiquitous in the terrestrial system with O and Si being the two most abundant elements in the Earth's crust (O 46.6% and Si 27.7%). Natural silica exists most commonly as quartz (the major constituent of sand) and in various living organisms. The ultrahigh background and the great diversity in silica family make it an extremely difficult task to distinguish the sources of SiO2 NPs in the environment[19].

Here, we report that it is possible to distinguish the sources of SiO2 NPs by their dual isotopic fingerprints. Si has three stable isotopes, $^{28}Si$, $^{29}Si$, and $^{30}Si$, with natural abundance 92.23%, 4.67%, and 3.10%. O also has three stable isotopes, $^{16}O$, $^{17}O$, and $^{18}O$, with $^{16}O$ being the most abundant (99.76%). Notably, studies on the biogeochemical cycle of Si and O revealed that different reservoirs of Si and O in the terrestrial system have different and limited ranges of Si and O isotopic compositions[20–22], suggesting that the isotopic fingerprints of natural silica should be constrained in certain ranges. The objective of this study is to explore whether the industrial synthetic processes of engineered SiO2 NPs lead to isotope fractionation of Si and O, which enables the differentiation of engineered NPs from their naturally occurring counterparts. This work demonstrates the feasibility of source distinguishing of NPs by their isotopic fingerprints, and therefore breaks through the past perception on inherent stable isotopic tracing of NPs. It also reveals some potential for distinguishing NPs from different manufacturers and synthesized by different synthetic methods, which should be important for analysis and monitoring of nano-products.

## Results

**Characterization of SiO2 NPs from different sources**. To test our hypothesis, we collected SiO2 NPs from a variety of sources with different properties and particle sizes (see Supplementary Fig. 1 and Supplementary Table 1). For natural SiO2 NPs, we selected two major forms of silica, quartz (NQ) and diatomite (ND), representing geologically and biologically originating silica ($n = 15$). For engineered SiO2 NPs, we collected SiO2 NP samples from different manufacturers located in different regions ($n = 50$; see Supplementary Table 2) synthesized by three dominating methods used in the industrial production, i.e., flame pyrolysis of silicon tetrachloride (SiCl4), precipitation of silicate solution, and sol-gel method[23–25]. The products of the three methods are called fumed silica (EF), precipitated silica (EP), and sol-gel silica (ES), respectively.

We first characterized SiO2 NPs from diverse sources by using traditional techniques. Figure 1a shows a SEM image of standard SiO2 NPs with a monodisperse spherical shape. However, engineered SiO2 NPs had irregular shapes with considerable agglomeration (Fig. 1b, c) due to less precise shape control in the industrial production. No evident difference in shape or agglomerating behavior of engineered SiO2 NPs were observed among different synthetic methods or different manufacturers (Supplementary Fig. 2). For natural quartz and diatomite, their intact particles had characteristic shapes, i.e., NQ particles had a crystal shape consisting of flat faces with specific orientations (Supplementary Fig. 3), and ND particles could maintain the special shape of dead diatoms (Fig. 1e). However, NQ and ND also contained a large amount of defected and fragmentary particles with irregular and unfeatured shapes (Fig. 1d, f). They might be produced by natural weathering or other physical processes[26] and were not characteristic enough to differ from engineered SiO2 NPs (Supplementary Fig. 3). Therefore, microscopy measurements only cannot distinguish the sources of SiO2 NPs.

The crystal structures and chemical compositions of SiO2 NPs were also characterized. The XRD yielded characteristic peaks of crystalline NQ but could not distinguish other amorphous SiO2 NPs (Fig. 1g). Furthermore, some natural and industrial processes may partially transform amorphous SiO2 NPs into crystal structures[27,28]. The EDX patterns showed that all SiO2 NP samples were comprised of only Si and O (Fig. 1h), and the atomic ratio of O to Si ($R_{O/Si}$) ranged from $1.42 \pm 0.56$ to $3.27 \pm 0.57$ (mean ± s.d.; Fig. 1i). It is interesting to note that the $R_{O/Si}$ of SiO2 NPs did not strictly equal to the stoichiometric ratio O:Si = 2:1. Most samples showed an excess of O, which could be attributed to the presence of silanol (-SiOH) groups and water (including structural and free water)[23]. Although engineered SiO2 NPs showed a slightly larger deviation from $R_{O/Si} = 2$ than naturally occurring ones, this difference was not sufficient to distinguish between them. Moreover, the $R_{O/Si}$ may change upon the removal of water or condensation of -SiOH groups[23], and it showed large variations at different locations even in the same sample (Supplementary Fig. 4). Overall, the currently available measures (i.e., by shape, crystal structure, or chemical composition) are not able to distinguish the sources of SiO2 NPs.

**Si and O isotopic signatures of SiO2 NPs**. We then determined the Si and O isotopic fingerprints of SiO2 NPs from diverse sources. Here, the isotopic composition of a sample is expressed

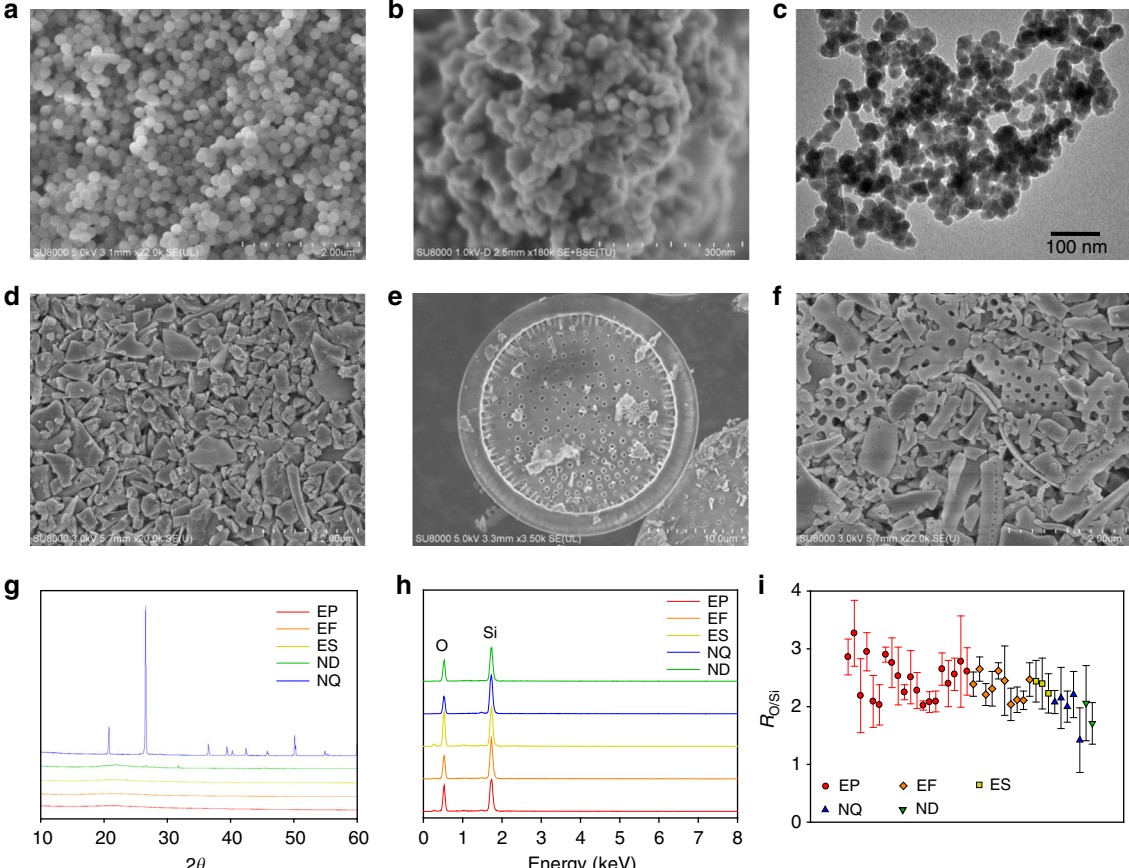

**Fig. 1** Characterization of SiO$_2$ NPs of different origins. **a** Typical SEM image of a SiO$_2$ NP standard. **b, c** Typical SEM (**b**) and TEM (**c**) images of engineered SiO$_2$ NPs. **d** SEM image of natural quartz particles. **e, f** SEM images of intact (**e**) and fragmentary (**f**) diatomite particles. **g**, XRD patterns of engineered and natural SiO$_2$ NPs. **h** EDX patterns of engineered and natural SiO$_2$ NPs. **i** The atomic ratio of O to Si ($R_{O/Si}$) of engineered and natural SiO$_2$ NPs based on the EDX measurements. The error bars represent 2s.d. ($n$ = 4-15). EP, EF, ES, ND, and NQ represent precipitated silica, fumed silica, sol–gel silica, diatomite, and quartz, respectively

as a $\delta$ value relative to a standard solution:

$$\delta^x E = \left( \frac{( {}^x E / {}^y E)_{\text{sample}}}{( {}^x E / {}^y E)_{\text{standard}}} - 1 \right) \times 1000‰ \qquad (1)$$

where $E$ represents an element ($E$ = Si or O), and $x$ and $y$ represent mass numbers of two isotopes of the element $E$ (the $y$ normally represents the mass number of the lightest stable isotope, i.e., $y$ = 28 for Si and 16 for O). The Si-O isotopic fingerprints are described by $\delta^{30}$Si and $\delta^{18}$O. High-precision Si isotope determination was achieved by multi-collector inductively coupled plasma mass spectrometry (MC-ICP-MS; 2s.d. = 0.3‰)[29].

The Si isotopic compositions of all samples followed the mass-dependent isotope fractionation (Supplementary Fig. 5). As shown in Fig. 2a, natural SiO$_2$ NPs showed narrow $\delta^{30}$Si ranges (-0.58–0.08‰ for NQ and 0.15–0.36‰ for ND), because the isotope fractionation of Si in the terrestrial system is greatly limited by its low volatility, chemical inertness, and invariant bonding environment (only form Si$^{4+}$)[30,31]. Noteworthily, we also compared our results with available data in the literature and found that the Si isotopic fingerprints of NQ and ND obtained here were highly consistent with those in the literature (Fig. 2b)[20,21,32–41]. This demonstrated a good representiveness of our samples. Figure 2b also indicates the geographical variations in $\delta^{30}$Si of NQ and ND. For engineered SiO$_2$ NPs, we found a significant negative shift in $\delta^{30}$Si (i.e., enriched in light isotope) from natural NPs ($P$ <10$^{-4}$). Especially, EF showed a broad $\delta^{30}$Si

range from −5.74‰ to −0.29‰ (Fig. 2a), with −5.74‰ approaching the most negative $\delta^{30}$Si value ever found in terrestrial samples[20].

Regarding O isotope, generally, the variations in $\delta^{18}$O were much larger than that in $\delta^{30}$Si due to higher chemical activity of O and larger difference in mass between $^{16}$O and $^{18}$O. From Fig. 2c, different sources of SiO$_2$ NPs also showed different $\delta^{18}$O ranges. NQ was $^{18}$O-depleted relative to ND, which also accorded with the previously published $\delta^{30}$Si data of NQ and ND (Fig. 2d)[34,41]. The $\delta^{18}$O of engineered SiO$_2$ NPs fell between that of NQ and ND ($P$<10$^{-4}$). Interestingly, EF showed a wider $\delta^{30}$Si range than ES and EP, but its $\delta^{18}$O range was narrower than that of ES and EP. The Si or O isotopic fingerprints of engineered SiO$_2$ NPs showed no evident trends with the particle size (Supplementary Fig. 6). From Fig. 2b, d, although different sources of SiO$_2$ NPs showed some difference in Si and O isotopic fingerprints, it was not able to fully distinguish the different sources by using Si or O isotopes alone due to partial overlapping of the isotopic distribution ranges. Thus, we further looked into the Si-O 2D isotopic fingerprints.

Figure 2e–h shows the $\delta^{30}$Si-$\delta^{18}$O plot of SiO$_2$ NPs. Different sources of SiO$_2$ NPs clustered into different zones. More importantly, NQ, ND, and engineered SiO$_2$ NPs (ES + EP + EF) could be fully differentiated into three isolated zones by two straight lines ($\delta^{18}$O = 13‰ and $\delta^{30}$Si = 0.1‰; Fig. 2e, f). This revealed the possibility of distinguishing engineered SiO$_2$ NPs from their naturally occurring counterparts by the $\delta^{30}$Si-$\delta^{18}$O

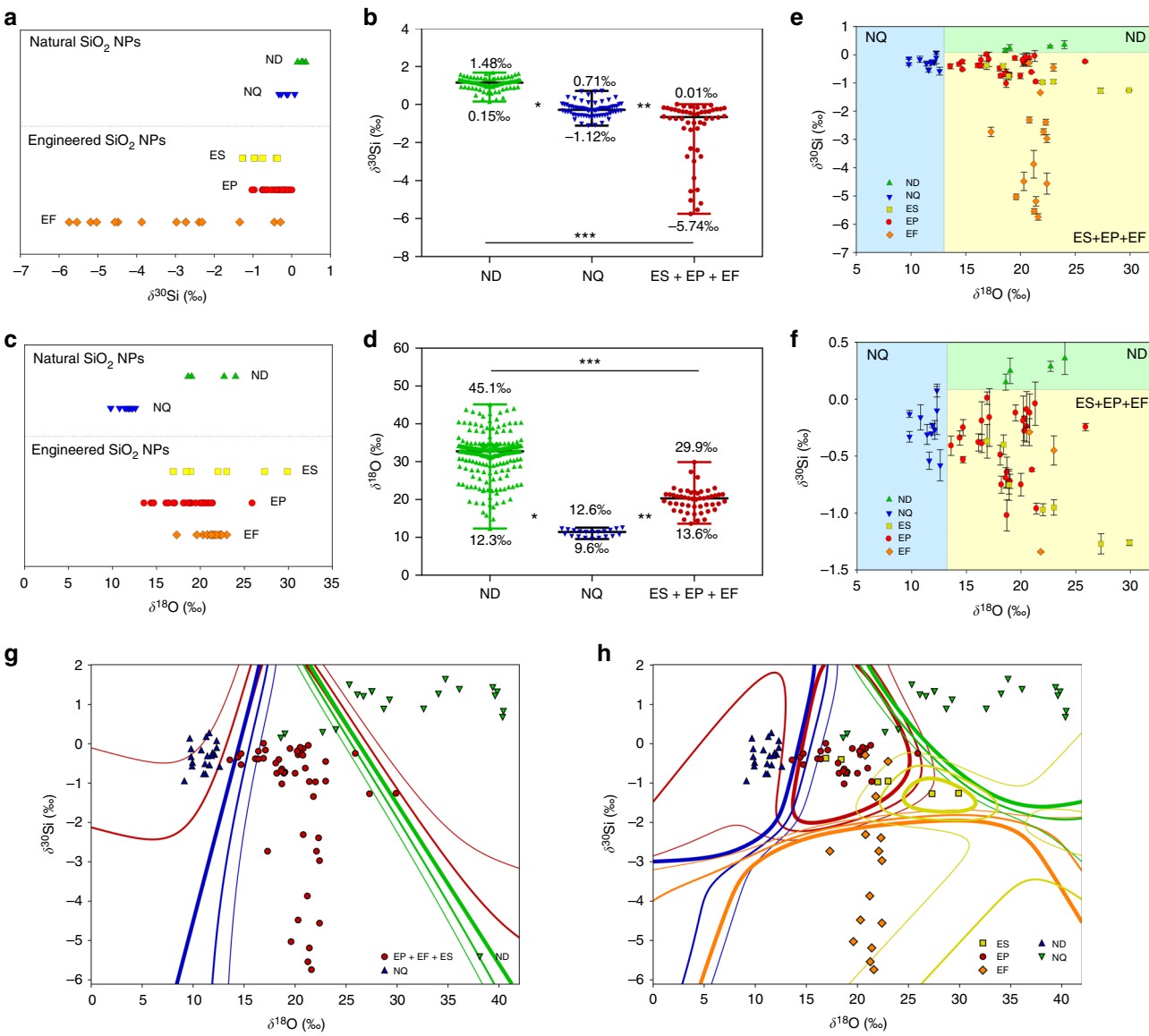

**Fig. 2** Si and O isotopic fingerprints of SiO$_2$ NPs of different origins. **a**, **b** Si isotopic composition of SiO$_2$ NPs grouped according to sources. **c**, **d** O isotopic composition of SiO$_2$ NPs grouped according to sources. The dots in **a** and **c** represent the samples tested in this study, and those in **b** and **d** include available data in the literature[20,21,32–41]. In **b** and **d**, *$P < 10^{-4}$, **$P < 10^{-4}$, and ***$P < 10^{-4}$. Unpaired Student's two-tailed t-test was used. **e**, **f**, Si-O 2D isotopic fingerprints of SiO$_2$ NPs with source differentiation by two straight lines ($\delta^{18}O = 13‰$ and $\delta^{30}Si = 0.1‰$). The **f** is a partial enlarged view of **e**. The different colored zones represent different sources (NQ or ND or ES+EF+EP). The error bars represent 2s.d. ($n = 2$-5). **g**, **h** Si-O 2D isotopic fingerprints of SiO$_2$ NPs with linear discriminant analysis (LDA) into three (**g**) or five classes (**h**). The zones defined by colored contour lines representing virtual distribution ranges of different sources given by the LDA-based classifiers. The color and thickness of the contour lines correspond to the respective sources and the probabilities of a sample being predicted to be the related class (0.5, 0.4, 0.3 for the thick, normal, and the thin one, respectively). Note that the ND and NQ samples in **g**, **h** include both the real samples used in this study and pseudo-samples constructed using the $\delta^{30}Si$ and $\delta^{18}O$ data reported in the literature (see Supplementary Methods 1.4)

isotopic fingerprints. To make the method more precise and quantitative, we developed a machine learning model to identify the source of a SiO$_2$ NP sample with linear discriminant analysis (LDA) into three, four, or five classes (see Methods). LDA is a supervised machine learning method that provides an efficient and accurate tool for multi-class classification problems[42]. The results for three and five classes are shown in Fig. 2g, h and that for four classes is given in Supplementary Fig. 7. The colored lines in Fig. 2g, h define the virtual distribution zones of different sources of SiO$_2$ NPs in 2D space constructed by two attributes, $\delta^{30}Si$ and $\delta^{18}O$, with different line thickness meaning different classification probability contours. Compared with the simple differentiation by two lines (Fig. 2e, f), the machine learning

model can theoretically enable the each source of SiO$_2$ NPs to be revealed, and it is also easy to be applied to other types of NPs with more than two elements. In this way, we have calculated the probabilities of sources for all SiO$_2$ NP samples (see Supplementary Tables 3-5), and the source discrimination results are given in Table 1 and Supplementary Table 6-7 based on the most probable source. The total discrimination accuracy between engineered and natural SiO$_2$ NPs was beyond 93.3%, indicating that this technique was highly accurate and reliable. Specifically, in terms of engineered NPs, the discrimination accuracy between EP and EF was > 80%, but ES could not be well differentiated from other sources. Therefore, this technique showed a strong ability of distinguishing between engineered and natural NPs,

while its potential to distinguish the different synthetic methods of engineered NPs needs to be further improved.

**Insights into the synthetic pathways of engineered SiO₂ NPs.** To better understand the difference in the isotopic fingerprints of SiO₂ NPs, we investigated the industrial synthetic pathways of engineered SiO₂ NPs. As shown in Fig. 3, the industrial production of SiO₂ NPs, including EF, ES, and EP pathways, involves complex chemical and physical processes. All chemical reactions and materials in the synthesis of engineered SiO₂ NPs are listed in Supplementary Table 8. Natural quartz is the most commonly used starting material (only in few cases diatomite is used), and there are five important intermediate substances, ferrosilicon ($Fe_xSi_y$), industrial Si, silicon tetrachloride ($SiCl_4$), tetraethoxysilane (TEOS), and sodium silicate ($Na_2SiO_3$). For Si isotopes, compared with the raw material NQ, all products (EF, ES, and EP) were enriched in the light Si isotope (Fig. 2a), which followed the kinetic isotope fractionation mechanism[8]. It should

be noted that the isotope fractionation caused by a reaction/process is dependent on the relative fraction reacted[8]. In industrial production, the highest possible relative fraction reacted is always pursued to achieve high yields, which can actually erase the isotope fractionation in products[8]. Comparing the routes to EF and ES (Fig. 3), EF showed a much more negative $\delta^{30}Si$ range than ES, suggesting that the particular reaction step of EF, i.e., the flame pyrolysis of $SiCl_4$ (reaction 10), might dominate the Si isotope variations of EF. This was also evidenced by the wide $\delta^{30}Si$ range of EF produced from the same precursor ($SiCl_4$). The wide $\delta^{30}Si$ range of EF could be explained by the large isotopic enrichment factor and highly uncertain relative fraction reacted of the reaction 10 (see Supplementary Section 2.1 for detailed discussion). ES and EP showed small shift in $\delta^{30}Si$ from NQ and limited $\delta^{30}Si$ ranges, suggesting that the sol-gel and precipitation processes caused only little isotope fractionation.

For O isotopes, in contrast to Si isotopes, all products (EF, ES, and EP) were enriched in the heavy isotope relative to NQ (Fig. 2c), which could not be explained by the kinetic isotope fractionation. Note that the raw material NQ was not the only source of O in the synthetic pathways. Thus, it is rational to infer that the enrichment of heavy O isotope in engineered SiO₂ NPs might result from the introduction of external $^{18}O$-enriched substances (e.g., $O_2$ in reaction 10, alcohol in reactions 8 and 9, and NaOH in reaction 5; see Supplementary Table 8). The industrial $O_2$ gas normally derives from atmospheric $O_2$ that is isotopically heavier ($24.15 \pm 0.05‰$[43]) than NQ, causing EF being enriched in $^{18}O$. For ES and EP, the explanation for their O isotope variations is still not very clear due to the complex O sources and unknown O isotopic compositions of industrial alcohol and NaOH, which needs to be verified in future studies. The uncertain O sources might also be the reason why ES and EP had wider $\delta^{18}O$ ranges than EF.

**Differentiation of engineered SiO₂ NPs according to manufacturers.** To further recognize the power of this technique, we classified the isotopic fingerprints of engineered SiO₂ NPs according to manufacturers (Fig. 4). Interestingly, we found that the engineered SiO₂ NPs from different manufacturers indeed showed some characteristic Si and O isotopic fingerprints. Especially for EP (Fig. 4a, b) and ES (Fig. 4c, d), the Si and O isotopic fingerprints of products from different manufacturers

---

**Table 1 Source discrimination results of SiO₂ NPs of known sources into five classes by the machine learning model[a]**

| Sample | Total | Source identified[b] | | | | | Accuracy |
|---|---|---|---|---|---|---|---|
| | | EP | EF | ES | NQ | ND | |
| SiO₂ NPs | 90 | Number of correct: 84[c] | | | | | 93.3% |
| └ Engineered NPs | 50 | 49[d] | | | 1[e] | | 98.0% |
| └ EP | 28 | 27 | 0 | 0 | 1 | 0 | 96.4% |
| └ EF | 15 | 3 | 12 | 0 | 0 | 0 | 80.0% |
| └ ES | 7 | 5 | 0 | 2 | 0 | 0 | 28.6% |
| └ Natural NPs | 40 | 5[d] | 35[e] | | | | 87.5% |
| └ NQ | 20 | 0 | 0 | 0 | 20 | 0 | 100% |
| └ ND | 20 | 5 | 0 | 0 | 0 | 15 | 75.0% |

[a] Engineered NPs were collected from 14 manufacturers located in 6 different regions. Natural NPs included both real and pseudo-samples (see Supplementary Section 1.4). For real samples, NQ samples were collected from 9 manufactures and ND samples were from 3 manufactures. More details about samples are given in Supplementary Table 1 and 2. The source discrimination results into three and four classes are given in Supplementary Table 6 and 7.
[b] The machine learning model could give a probability value for each candidate source (see Supplementary Table 3), and the statistics in this table was based on the most probable source.
[c] The "number of correct" means the number of samples with correct discrimination result between engineered and natural SiO₂ NP.
[d] The total number of engineered SiO₂ NPs identified (EP + EF + ES).
[e] The total number of natural SiO₂ NPs identified (NQ + ND).

---

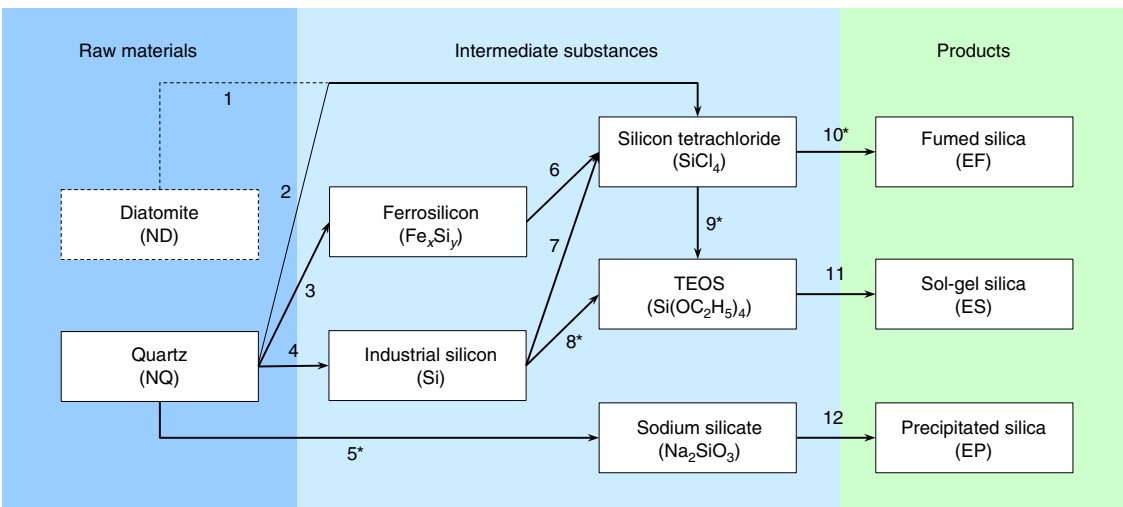

**Fig. 3** Scheme showing the industrial synthetic pathways of engineered SiO₂ NPs. The reaction equations **1-12** are given in Supplementary Table 8. The key reactions that may cause significant isotope fractionation of Si or O are marked with asterisks. The dashed line represent a potential but not commonly used route

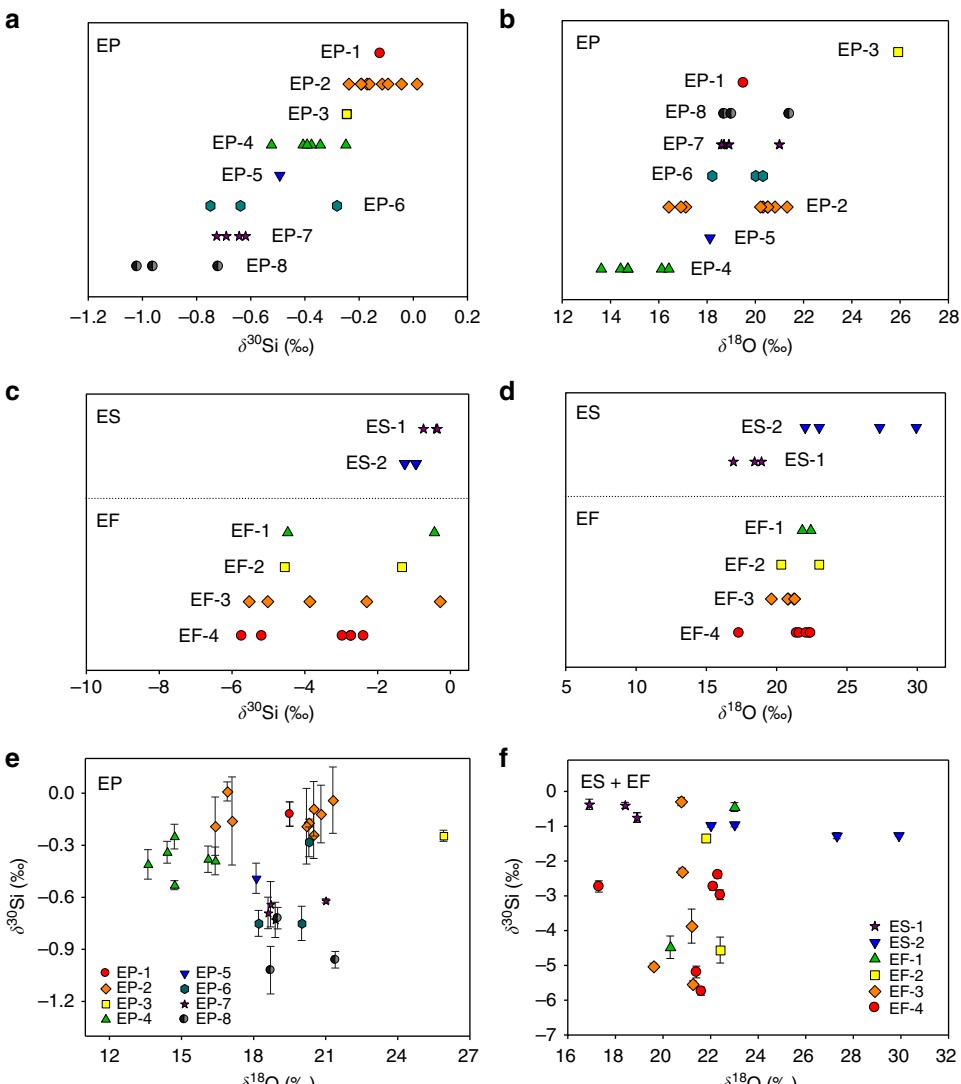

**Fig. 4** Si and O isotopic fingerprints of engineered $SiO_2$ NPs synthesized by different methods and from different manufacturers. **a**, **b** Si (**a**) and O (**b**) isotopic compositions of precipitated $SiO_2$ NPs (EP) grouped according to manufacturers. **c**, **d** Si (**c**) and O (**d**) isotopic compositions of sol–gel and fumed $SiO_2$ NPs (ES and EF) grouped according to manufacturers. **e** Si-O 2D isotopic fingerprints of EP. **f** Si-O 2D isotopic fingerprints of ES and EF. The error bars in e and f represent 2s.d. in parallel measurements ($n = 2$–5)

distributed in different ranges, probably resulting from the variation in Si and O isotopic composition of the raw material. This enabled the potential differentiation of EP and ES products from different manufacturers by their Si-O 2D isotopic fingerprints (Fig. 4e, f). For EF, the products from different manufacturers also showed some difference in Si and O isotopic fingerprints (Fig. 4c, d), but it was not large enough to differentiate among different manufacturers (Fig. 4f). Furthermore, we found that the isotope fractionation degree during the manufacturing process varied among different manufacturers (Supplementary Fig. 8), suggesting that the isotope fractionation was also affected by the manufacturing conditions. Overall, this technique is not only capable of differentiating the sources of $SiO_2$ NPs, but also shows some capability to identify their manufacturers.

**Application to real samples.** We finally applied this technique to analyze consumer products that claimed to contain $SiO_2$ NPs, including several types of toothpastes (TP), inorganic filter membranes (IFM), and nanoquartz coating (NQC). We extracted $SiO_2$ NPs from these consumer products (Supplementary

Table 9), measured their Si-O isotopic fingerprints, and calculated their probabilities of sources using the three-, four-, and five-class LDA-based classifiers (Supplementary Fig. 9 and Supplementary Table 10-12). It was found that the $SiO_2$ NPs in TP samples were highly probable to originate from EP or ES (probability 90.0–96.0% for EP + ES in four-class LDA), which accorded with the fact that precipitated silica is commonly used as an abrasive and thickening agent in toothpastes due to its abrasive nature. The $SiO_2$ NPs in IFM samples probably originated from EP or NQ (probability 52.5–58.4% for EP and 37.1–43.9% for NQ in five-class LDA), and that in the NQC sample most probably came from NQ (probability > 59.5%), which was consistent with the production description provided by the factory. These results showed the usefulness of the technique in real sample analysis.

## Discussion
Our results have revealed the possibility of isotopic fingerprints in source tracing of NPs, which actually breaks through the previous knowledge on stable isotopic tracing of NPs[11,12]. It should be

stressed that this technique is based on the inherent isotopic fingerprints of NPs and thus should be suitable for application in complex systems (e.g., natural environment, biological, and industrial systems). The difference in isotopic fingerprints of different sources of $SiO_2$ NPs are deemed to result from the Si and O isotope fractionation during the manufacturing process of engineered $SiO_2$ NPs as well as the different isotopic compositions of raw materials. The combined use of Si and O isotopic signatures provide more information on the sources of NPs, enabling a more effective source differentiation than using isotopes of one single element. Note that most types of engineered NPs (e.g., $TiO_2$, $Fe_xO_y$, $ZrO_2$, quantum dots) have multiple elements with multiple isotopes. As long as their naturally occurring counterparts have relatively constant isotopic ranges and that the manufacturing process of engineered NPs leads to a stable isotope fractionation, it is possible to differentiate the sources of NPs by their isotopic fingerprints. Considering that many elements have constrained isotopic composition ranges in the terrestrial system, this technique has the potential to emerge as a universal tool for source distinguishing of NPs.

The correct distinguishing of sources of NPs in samples is an important prerequisite for a proper risk assessment of engineered NPs. Although the environmental concentrations of engineered NPs are currently very low compared with their naturally occurring counterparts, an exponential increase is predicted due to their rising usage and disposal amounts[44,45]. The distinguishable isotopic fingerprints reveal a possible approach to identify whether the target NP samples are anthropogenic or naturally occurring for properly assessing the impact of NP exposure. Furthermore, the potential of distinguishing the manufacturer and synthetic methods of engineered NPs, which is very difficult to be accomplished by other techniques, would be of high value for analysis and monitoring of nano-products.

By far, the shortage of this technique is that, due to the great diversity of silica family, it does not cover some rare types of $SiO_2$ NPs; either, it is difficult to predict whether the future technical improvement in the production of engineered $SiO_2$ NPs will significantly alter their isotopic fingerprints. With regards to more types of NPs, for some elements it is still difficult to precisely measure their stable isotopic compositions[46]. For real-world samples, intensive sample purification is required prior to high-precision stable isotopic analysis. Therefore, this technique still needs continuous improvement in future applications. Future works will be needed to: (1) get deeper insights into the isotope fractionation mechanisms during the natural and engineering processes of NPs, (2) further optimize the mathematical model by including more types of sources and a larger size of sample set to make it more accurate and practical, and (3) extend the technique to more types of NPs.

## Methods

**Characterization of $SiO_2$ NPs.** SEM images were capture on a Hitachi S-3000N scanning electron microscope (Tokyo, Japan) equipped with an energy dispersive X-ray spectroscope operating at an accelerating voltage of 15kV. TEM measurements were performed on a 300-mesh copper grid using a JEM 2100 transmission electron microscope (JEOL, Japan) operating at 200kV. XRD analyses were performed using a PANalytical X'Pert X-ray diffractometer (Almelo, Netherlands) at a scanning rate of 10°/min. XRF analyses were performed on an ARL Perform'X X-ray fluoroscope (ThermoFisher, Switzerland). The elemental concentrations were measured by an Agilent 7500 inductively coupled plasma mass spectrometer (Santa Clara, CA, USA).

**Sample preparation for Si isotopic analysis.** For Si isotopic analysis, the sample was digested using alkali fusion method with solid NaOH[47,48]. Briefly, 5-10 mg of powdered samples were mixed with 200mg of solid NaOH in a 30mL Ag crucible, and then the mixture was heated at 1000K in a muffle furnace. After cooling down to room temperature, the fusion cake was dissolved with 10 mL of water followed by being stored in dark for 24h. The final solution was transferred to a 50mL

centrifuge tube, and then its pH was adjusted to ~2 using HCl solution. The blank Ag crucible and solid NaOH were also analyzed using the same procedures to ensure that they had no interference to the measurement of Si isotopic composition.

To eliminate the interference from sample matrix, cation-exchange chromatography was employed to purify the sample as reported previously[47,49]. The cation-exchange resin (DOWEX 50-X12, 200-400 mesh) was first activated for 12h in dark and packed to a 1.8mL resin bed. Then, the resin was repeatedly rinsed with HCl and $HNO_3$ solution followed by being eluted to neutral pH with water. Afterwards, 2mL of sample at a Si concentration of 2mg/L was loaded to the cation-exchange column and eluted with 2mL of water. The final solution can be directly analyzed by multi-collector inductively coupled plasma mass spectrometry (MC-ICP-MS). The recovery of Si during the sample preparation process was >95%. The whole sample preparation procedures were tested with two Si isotope standard reference materials (NIST SRM-8546 and IRMM-017) to verify that no interference was caused to the Si isotope measurement.

**Si isotopic analysis.** The Si isotopic composition was measured by a Nu Plasma II MC-ICP-MS (Wrexham, UK) equipped with a DeSolvation Nebulizer System (DSN-100). Instrumental sensitivity was ~6.7Vμg$^{-1}$g for $^{28}$Si in medium-resolution mode. The sample was introduced in dry mode using a PFA nebulizer at a flow rate of 70μL/min with a signal intensity of $^{28}$Si in the range of 3.6–7.5V. The optimized instrumental parameters are listed in Supplementary Table 13. The signal intensities of blank HCl and NaOH pretreated using the same procedures as mentioned above were less than 0.04V, indicating that they caused no interference to the Si isotope ratio measurements. After each measurement, a rinse with HCl solution (pH = 2) for >120s was used to reduce the background signal intensity to < 0.03V. At least two parallel measurements were performed for all samples.

The mass bias was corrected by the standard-sample-standard bracketing method. The Si isotope composition in a sample is expressed by a δ value ($δ^{30}$Si and $δ^{29}$Si) relative to the standard NIST SRM-8546:

$$\delta^{29}Si = \left( \frac{\left(^{29}Si/^{28}Si\right)_{sample}}{\left(^{29}Si/^{28}Si\right)_{standard}} - 1 \right) \times 1000‰ \quad (2)$$

$$\delta^{30}Si = \left( \frac{\left(^{30}Si/^{28}Si\right)_{sample}}{\left(^{30}Si/^{28}Si\right)_{standard}} - 1 \right) \times 1000‰ \quad (3)$$

Two standard reference materials (NIST SRM-8546 and IRMM-017) were used to validate the method. In each sample batch, the difference of signal intensity between standard and sample solutions was < 10%. A $δ^{30}$Si value of $-(0.004 \pm 0.17)$‰ (mean ± s.d., $n = 27$) was obtained with a NIST SRM-8546 solution, and the $δ^{30}$Si value of IRMM-017 was $-(1.43 \pm 0.16)$‰ (mean ± s.d., $n = 22$), which was very close to the previously reported results[29,48], proving that our method was highly accurate and precise.

**O isotopic analysis.** O isotopic ratios were measured by the bromine pentafluoride method[50]. Briefly, to liberate oxygen from $SiO_2$, $BrF_5$ was used to react with the sample under high vacuum ($< 2\times 10^{-3}$Pa) and high temperature (550 °C) for more than six hours. Then, the product ($O_2$) collected by a sample hose with a 5 Å molecular sieve was directly subjected to O isotopic ratio measurement using a Thermo 253 Plus isotope ratio mass spectrometer (IR-MS). In each batch, a standard reference material (GBW04421) was inserted into five samples to verify that no O isotope fractionation occurred in this process. The O isotopic composition in a sample was expressed by a $δ^{18}$O value:

$$\delta^{18}O_{sample-standard} = \left( \frac{\left(^{18}O/^{16}O\right)_{sample}}{\left(^{18}O/^{16}O\right)_{standard}} - 1 \right) \times 1000‰ \quad (4)$$

To facilitate inter-laboratory data comparison, the O isotope ratios are also usually reported relative to the "Vienna Standard Mean Ocean Water" (VSMOW). The calculation formula is as follows:

$$\delta^{18}O_{SA-VSMOW} = \frac{\left(\delta^{18}O_{SA-RE} + 1000\right)\left(\delta^{18}O_{ST-VSMOW} + 1000\right)}{\left(\delta^{18}O_{ST-RE} + 1000\right)} - 1000(‰) \quad (5)$$

where SA represents the sample, ST represents the standard reference material (GBW04421), and RE represents the reference gas ($O_2$) used in IR-MS. The $δ^{30}$Si value of GBW04421 was (10.92 ± 0.32)‰ (mean ± s.d., $n = 13$) relative to VSMOW.

**Machine learning model.** In order to mathematically distinguish the potential sources of $SiO_2$ NPs by isotopic fingerprints, we built classifiers with linear discriminant analysis (LDA). LDA is a supervised machine learning method that provides an efficient and accurate tool for multi-class classification problems[42]. The experimental isotopic fingerprint data for the three to five classes of sources, combining with the additional literature data for NQ and ND (Supplementary Section 1.4), formed the training set. Data preparation and analysis were performed using our in-house Python scripts, and the LDA implementation was based on the scikit-learn (0.19.0) package[51].

**Reporting Summary**. Further information on experimental design is available in the Nature Research Reporting Summary linked to this article.

## Data availability
The data that support the findings of this study are available from the corresponding author upon reasonable request.

## Code availability
The code that support the findings of this study are available from the corresponding author upon reasonable request.

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

## Acknowledgements

This work was financially supported by the National Natural Science Foundation of China (21825403, 91843301, 91543104, 91743204), the Chinese Academy of Sciences (XDB14010400, QYZDB-SSW-DQC018), and the National Basic Research Program of China (2015CB931903, 2015CB932003). Q. L. acknowledges the China National Special Support Program for Young Top-Notch Talents.

## Author contributions

Q. L. designed the research; G. J. supervised the project; X. Y. performed most of experiments; X. L. and A. Z. built the machine learning model for source identification; D. L. helped with the isotopic analysis; G. L. helped with the XRD measurements; Q. Z. gave comments on the paper; Q. L. and X. Y. analyzed the data; Q. L., X. Y., and A. Z. wrote the paper.

## Additional information

**Competing interests:** The authors declare no competing interests.

**Journal Peer Review Information**: *Nature Communications* thanks Frank Vanhaecke, and the other anonymous reviewers for their contribution to the peer review of this work. Peer reviewer reports are available

