## [Peer Review File · Nature Communications]

Reviewers' comments:

Reviewer #1 (Remarks to the Author):

In this manuscript, the authors demonstrate differences in the isotopic composition of both Si and O between SiO₂ nanoparticles (NPs) of different origin. It is shown that – at least to some extent – taking into account both isotopic signatures allows one to distinguish between natural and synthetic (or engineered) nanoparticles and between engineered particles manufactured via different approaches. Basically, this is a very interesting finding and therefore, I would like to recommend publication of this manuscript, either in NCOMMS or in a journal devoted to analytical chemistry. However, I think that, to some extent, the authors oversell the capabilities of the approach a bit, while it is also a pity that some hypotheses were not experimentally tested.

Major remarks

-I do agree that the method described is promising for distinguishing between natural and engineered NPs and to distinguish between engineered NPs manufactured via different approaches (to some extent), but I fail to see how the method could contribute to nanotechnology development and/or nanoprodukt quality management?

-The major conclusion the authors draw from the section 'Application to real samples' is that "these results demonstrate the applicability of the technique in real sample analysis". I tend to disagree as the validity (correctness) of the results obtained / conclusions drawn has not been demonstrated. Or do the authors know the nature of the SiO₂ nanoparticles in the consumer products investigated?

-The authors hypothetically attribute the isotopically heavy O in engineered SiO₂ nanoparticles (ES & EP) to the use of, among other, alcohol or NaOH with a higher fraction of ¹⁸O. It is regrettable that this hypothesis has not been experimentally validated.

-Did the authors ensure that fusion at 1000 K does not give rise to Si isotope fractionation? Was a reference material with known Si isotopic composition submitted to the same sample pre-treatment?

-In my opinion, Table 1 demonstrates that the conclusions should be formulated more prudently. "ES" SiO₂ nanoparticles are, e.g., largely misidentified. The authors do phrase it more prudently in lines 165-166, but on some other locations, they seem to be a bit overenthusiastic.

-Could the authors indicate what the geographical variation of the natural quartz / diatom SiO₂ nanoparticles is? Or could they explicitly mention what the global range of variation in δSi is for these materials?

-Table 2 is not entirely clear. For information on the number of samples N, the supplementary material needs to be consulted. Therefore, I would like to recommend somewhat more information on "N" (# of manufacturers, # of batches, ...) to be included in the main paper. Could it be that the value '15' of correctly identified natural NPs should be '35' instead?

-I am not entirely convinced by Fig. S6 (supplementary information). While it is indicated that the NPs come from a single manufacturer, there is an 8 permil difference in δ¹⁸O value between the samples investigated, while the δ³⁰Si value is more or less stable. Can this be explained based on the manufacturing process used by this manufacturer? Is this sufficient evidence (the lack of a systematic trend) that the NP size does not affect the isotopic composition?

-Table S7 indicates that the voltage for 28Si varied between 3.6 and 7.5 V. As the analyte concentration / signal intensity has an effect on the extent of mass discrimination, I wonder if this refers to intra-day or inter-day variation? Could the authors more explicitly indicate what range of sensitivity / analyte concentration they allowed within a measurement series? How close was the concentration matching between the sample and standard solutions?

-I would like to recommend to include Supplementary figure S7 in the main article as it nicely demonstrates the capability of distinguishing between engineered and natural SiO₂ nanoparticles.

Minor remarks

-I don't know whether a method relying on 2 isotope ratios or isotopic signatures should be called "multi-isotopic" as the latter seems to suggest a higher number.

-Although the authors have a reference to the 2018 Gondikas et al. paper in Environmental Science – nano, they fail to indicate that multi-element analysis of TiO₂ NPs using a single- particle ICP-MS approach with a Time-of-Flight (ToF) ICP-MS instrument provided a basis for distinguishing between natural and engineered NPs.

-Lines 228-229. I suggest the information on labeling/ tracers to be removed.

-Lines 293-294. The desolvation system was coupled to the MC-ICP-MS instrument and not vice versa.

-Line 313. A unit (K?) is missing. The same is true on a couple of locations in the supplementary material.

-Line 317. I would rather say that the standard reference material was used to verify that no isotope fractionation occurred.

-Table S7. Some more information on the sampling cone / skimmer types is required. Characteristics? Goals (e.g., higher sensitivity, higher stability, ...)?

Reviewer #2 (Remarks to the Author):

I found the article 'Distinguishing the sources of silica nanoparticles by natural multi-isotopic fingerprints' to be very well written, the presented data was extensive and largely backed up the discussion. Machine learning approaches for this type of analyses are gaining increased attention and have a lot of potential benefits to offer. I believe the article is original and would be a valuable contribution both to the field of nanoanalytics and machine learning. My main concern with the manuscript is that the authors could be clearer about the limitations of the current model they have developed. I find the claims that the authors can distinguish ES from EP and that using stable isotope composition for source tracing in complex samples are not justified based on the data presented and may need to be modified. I have detailed my specific comments and suggestions below:

Line 35 -36: 'However, it is still one of the most challenging tasks in nanotechnology due to high similarity in physico-chemical properties between naturally occurring and engineered NPs.' – suggest rephrasing. It is often the novel physico-chemical properties of engineered nanomaterials that make them useful, for many engineered nanoparticles – Ag, Au, Cu, Ce etc it is the high purity of the metal

that makes them distinguishable from natural particles – as is the case in reference 3 from the previous sentence. However this still does not make them easy to detect/distinguish in natural media.

Line 220: 'It was found that the SiO₂ NPs in TP samples were highly probable to originate from EP (probability 53.3–79.9%),' – Supplementary table 3 shows that 5 of the 7 ES samples and 24 of the 28 EP samples fall in this probability range. Thus from this and figures 2 and S7 it seems that ES and EP are indistinguishable and the silica in the toothpaste could be either ES or EP.

Line 229-230: 'This makes it essentially different from currently used isotopic labelling methods⁷ and thus suitable for source tracing in complex systems (e.g., natural environment, biological, and industrial systems).'

&

Line 247 – 249: 'The distinguishable isotopic fingerprints provide a practical approach for label-free detection of engineered NPs against high background signals in the natural environment.' – I do not think this is a valid claim based on the data presented in this paper. There is a strong argument to be made that, when a sample is 100% natural or engineered, the isotope signature may be used to distinguish between the two. However considering the variability in isotope composition of both engineered and natural sources and the magnitude of the differences it seems highly unlikely that this would be possible. It is also likely that the relative proportion of engineered to total silica in a natural environmental sample will be too small to result in difference that is resolvable from the natural variability. These claims either need to be toned down or justified by theoretical mixing calculations or analysis of environmental samples.

Table 1: - the number of correctly identified natural NPs should be 35 not 15.

Figure 2e – This represents the main output of the presented work and is essentially the same data as that in supplementary figure 7, with a few additional data points. In figure S7 the graph is split into three fields – two natural and one for engineered. Have the authors considered including these zones/lines in figure 2e? Using the criteria in figure 7 there is a 100% success in distinguishing between natural and engineered and only 93.3% when using the machine learning approach so is this arguably more reliable than the LDA? From figure 2e it is apparent that sol-gel and precipitated silica are indistinguishable, and hence why the LDA identifies 5 of the 7 sol-gel samples as precipitated silica and the resulting contours for sol-gel silica do not seem logical. This may be because the model is trying to force a difference where none exists. Have the authors performed the LDA by with different groups – NQ – ND – ES+EP+EF and/or NQ – ND – EF – ES+EP?

As a side comment I would encourage the authors to consider how this plot might be made easier to read for someone who is colour-blind – for example using different shapes for data points instead of just different colours.

Figure 4 caption Lines 520, 521, 522 – consider removing 'natural' from 'natural Si and O isotopic' To remove any possible confusion as you refer to natural silica and engineered silica.

Response to Reviewer Comments

Response to Reviewer #1:

Response to major remarks:

1. **Question:** I do agree that the method described is promising for distinguishing between natural and engineered NPs and to distinguish between engineered NPs manufactured via different approaches (to some extent), but I fail to see how the method could contribute to nanotechnology development and/or nanoproduct quality management?

Answer: Thanks for your comment. We believe that source distinguishing of NPs is of high importance for nanotechnology risk assessment and nano-product analysis and related fields. To avoid any misunderstanding or overstatements, the related statements have been revised to be more prudent and accurate throughout the manuscript, e.g., “nanotechnology development” has been revised to “nano research”, “nano-product quality management” has been revised to “nano-product analysis”, etc. Please see page 3 line 32-34, page 5 line 75, page 13 line 259, etc.

2. **Question:** The major conclusion the authors draw from the section ‘Application to real samples’ is that “these results demonstrate the applicability of the technique in real sample analysis”. I tend to disagree as the validity (correctness) of the results obtained / conclusions drawn has not been demonstrated. Or do the authors know the nature of the SiO₂ nanoparticles in the consumer products investigated?

Answer: Thanks for your question. The section “Application to real samples” has two aims: The first aim is to report analytical results of some real products. These consumer products were directly purchased from local superstores. Although we did not fully know the exact sources of NPs in these products, some facts partly supported our results. For example, we found that SiO₂ NPs in TP samples were highly probable to originate from EP or ES, which accorded with the fact that precipitated silica is one of the most commonly used abrasive agents in toothpastes; the SiO₂ NPs in the NQC sample most probably came from NQ, which was also consistent with the production description provided by the manufacturer.

Furthermore, another aim of this section is to include the sample preparation procedures for real samples (as described in Supplementary Section 1.3). The effectiveness of the sample preparation procedures has been validated by the high purities of SiO₂ in the extracts (please see Supplementary Table 9).

Finally, we have mitigated the conclusion in this section by revising “demonstrated the applicability of the technique” to “showed the usefulness of the technique” to make it more correct. Please see page 12 line 232.

3. **Question:** The authors hypothetically attribute the isotopically heavy O in engineered SiO₂ nanoparticles (ES & EP) to the use of, among other, alcohol or NaOH with a higher fraction of ¹⁸O. It is regrettable that this hypothesis has not been experimentally validated.

Answer: Thanks for your suggestion. We quite agree that it is meaningful to experimentally validate whether industrial alcohol or NaOH are enriched in ¹⁸O. Unfortunately, up to now, there has been no available method capable of measuring the O isotopic composition in alcohol or NaOH. As a result, to the best of our knowledge, there has been no O isotope data of alcohol or NaOH reported in the literature. Thus, this hypothesis can only be verified in future studies. This point has been specified in the revised manuscript. Please see page 10 line 202-203.

On the other hand, in all three synthetic routes (EF, ES, and EP), the engineered SiO₂ NPs were enriched in heavy O isotope, which was different from that of Si. Considering that the Si followed the kinetic isotope fractionation mechanism, we could not simply ascribe the enrichment of ¹⁸O to equilibrium isotope fractionation. Furthermore, the potential sources of O in engineered SiO₂ NPs were much more complex than that of Si (see Supplementary Table 8). Therefore, although we could not identify the exact source for the enrichment of ¹⁸O in engineered SiO₂ NPs, it should be safe to infer that it might result from the introduction of external ¹⁸O-enriched substances.

4. **Question:** Did the authors ensure that fusion at 1000 K does not give rise to Si isotope fractionation? Was a reference material with known Si isotopic composition submitted to the same sample pre-treatment?

Answer: We have tested the whole sample preparation procedures (including the step of fusion at 1000 K) with two Si isotope standard reference materials (NIST SRM-8546 and IRMM-017) to verify that they caused no interference to the Si isotope measurement. This point has been noted in the revised paper. Please see page 15 line 296-298. This sample preparation method has also been adopted in previous studies (e.g., *Chem. Geol.* 2006, 235, 95-104; *J. Anal. At. Spectrom.* 2006, 21, 734-742; *Environ. Sci. Technol.* 2018, 52, 1088-1095).

5. **Question:** In my opinion, Table 1 demonstrates that the conclusions should be formulated more prudently. “ES” SiO₂ nanoparticles are, e.g., largely misidentified. The authors do phrase it more prudently in lines 165-166, but on some other locations, they seem to be a bit overenthusiastic.

Answer: Thanks for your suggestion. We have carefully checked the related statements all through the manuscript to ensure that the distinguishing capability for EF, ES, and EP is properly described. Please see page 5 line 73, page 11 line 211 and 226-227, page 13 line 257, and Supplementary Fig. 9. Furthermore, in this revision, we have added the three- and

four-class LDA-based classification results. This should help better understand the capability of this technique.

6. **Question:** Could the authors indicate what the geographical variation of the natural quartz / diatom SiO₂ nanoparticles is? Or could they explicitly mention what the global range of variation in δSi is for these materials?

Answer: The geographical variations of Si and O isotopic compositions in NQ and ND have actually been explicitly labeled in Fig. 2b and 2d, which include available data in the literature.

7. **Question:** Table 1 is not entirely clear. For information on the number of samples N , the supplementary material needs to be consulted. Therefore, I would like to recommend somewhat more information on “ N ” (# of manufacturers, # of batches, ...) to be included in the main paper. Could it be that the value ‘15’ of correctly identified natural NPs should be ‘35’ instead?

Answer: Thanks for your suggestion. We have added more information on the samples N in the footnote of Table 1. Please see page 25 line 489-493. In addition, the wrong number “15” in Table 1 has been corrected to “35”.

8. **Question:** I am not entirely convinced by Fig. S6 (supplementary information). While it is indicated that the NPs come from a single manufacturer, there is an 8 permil difference in $\delta^{18}\text{O}$ value between the samples investigated, while the $\delta^{30}\text{Si}$ value is more or less stable. Can this be explained based on the manufacturing process used by this manufacturer? Is this sufficient evidence (the lack of a systematic trend) that the NP size does not affect the isotopic composition?

Answer: Thanks for your comment. We have added more sample data in the Supplementary Fig. 6 covering all three different synthetic methods (EP, ES, and EF). For a certain method, the samples were from a same manufacturer. As shown in Supplementary Fig. 6, for all synthetic methods, the Si-O isotopic fingerprints of engineered SiO₂ NPs showed no clear trends with their particle size. Thus, this phenomenon should not be caused by a specific manufacturing process. Furthermore, we agree that it may be premature to conclude that the NP size did not affect the isotopic composition. So, the related statement has been revised to “The Si or O isotopic fingerprints of engineered SiO₂ NPs showed no evident trends with the particle size”. Please see page 8 line 142-143 and the caption of Supplementary Fig. 6.

9. **Question:** Table S7 indicates that the voltage for ^{28}Si varied between 3.6 and 7.5 V. As the analyte concentration / signal intensity has an effect on the extent of mass discrimination, I wonder if this refers to intra-day or inter-day variation? Could the authors more explicitly indicate what range of sensitivity / analyte concentration they allowed within a measurement

series? How close was the concentration matching between the sample and standard solutions?

Answer: Thanks for your question. The samples in this study were measured in three batches at different injection concentrations (depending on the original Si concentration in samples). The instrumental sensitivity was $\sim 6.7 \text{ V } \mu\text{g}^{-1} \text{ g}$ for ^{28}Si , and the signal intensities of all samples were in the range of 3.6-7.5 V (please see page 15 line 301-303). In each batch, the samples were diluted to the same Si concentration as standard solution (NIST8546). The difference in signal intensity between standard solution and samples in a batch were $< 10\%$, and both intra-day and inter-day variations of signal intensity were $< 20\%$. Please see page 16 line 315-316.

10. **Question:** I would like to recommend to include Supplementary figure S7 in the main article as it nicely demonstrates the capability of distinguishing between engineered and natural SiO_2 nanoparticles.

Answer: Thanks for your suggestion. We have included the old Supplementary Fig. 7 in Fig. 2 (now as Fig. 2e and 2f) in the revised manuscript.

Response to minor remarks:

1. **Question:** I don't know whether a method relying on 2 isotope ratios or isotopic signatures should be called "multi-isotopic" as the latter seems to suggest a higher number.

Answer: Thanks for your suggestion. We have carefully considered the use of "multi-". According to several dictionaries (Oxford, Longman, Collins, and Merriam-Webster), the prefix "multi-" can mean "more than one" (e.g., multiparous, multi-photon microscope). More importantly, this term highlights the difference from "single" element used in previous studies and also keeps the flexibility to be used in NP systems with more than two elements. For these considerations, we would like to retain the use of "multi-". Anyway, if you still think "multi-" is not suitable in this case, we are very willing to use other words.

2. **Question:** Although the authors have a reference to the 2018 Gondikas et al. paper in Environmental Science - nano, they fail to indicate that multi-element analysis of TiO_2 NPs using a single-particle ICP-MS approach with a Time-of-Flight (ToF) ICP-MS instrument provided a basis for distinguishing between natural and engineered NPs.

Answer: Thanks for your suggestion. The work by Gondikas et al. on multi-element analysis of TiO_2 NPs using spICP-MS and spICP-TOF MS has been indicated more clearly in the Introduction section. Please see page 3 line 37-39.

3. **Question:** Lines 228-229. I suggest the information on labeling/ tracers to be removed.

Answer: The information on labeling/tracers has been removed. Please see page 12 line 238.

4. **Question:** Lines 293-294. The desolvation system was coupled to the MC-ICP-MS

instrument and not vice versa.

Answer: In page 15 line 300, the “coupled to” has been revised to “equipped with”.

5. **Question:** *Line 313. A unit (K?) is missing. The same is true on a couple of locations in the supplementary material.*

Answer: The missing unit (°C) has been added in proper places.

6. **Question:** *Line 317. I would rather say that the standard reference material was used to verify that no isotope fractionation occurred.*

Answer: In page 16 line 325, the “ensure” has been changed to “verify”.

7. **Question:** *Table S7. Some more information on the sampling cone / skimmer types is required. Characteristics? Goals (e.g., higher sensitivity, higher stability, ...)?*

Answer: A sampler cone and a skimmer cone made of nickel specially for dry mode were used, since all samples were measured in dry mode by Nu Plasma II MC-ICP-MS equipped with a DSN system. This point has been noted in Supplementary Table 13.

Response to Reviewer #2:

1. **Question:** *Line 35-36: ‘However, it is still one of the most challenging tasks in nanotechnology due to high similarity in physico-chemical properties between naturally occurring and engineered NPs.’ – suggest rephrasing. It is often the novel physico-chemical properties of engineered nanomaterials that make them useful, for many engineered nanoparticles – Ag, Au, Cu, Ce etc it is the high purity of the metal that makes them distinguishable from natural particles – as is the case in reference 3 from the previous sentence. However this still does not make them easy to detect/distinguish in natural media.*

Answer: Thanks for your suggestion. The related statement has been rephrased to “Although engineered NPs are usually produced in high purity, it is still one of the most challenging tasks in nanoanalytics to detect/distinguish them in complex natural media.” Please see page 3 line 34-36.

2. **Question:** *Line 220: ‘It was found that the SiO₂ NPs in TP samples were highly probable to originate from EP (probability 53.3–79.9%),’ – Supplementary table 3 shows that 5 of the 7 ES samples and 24 of the 28 EP samples fall in this probability range. Thus from this and figures 2 and S7 it seems that ES and EP are indistinguishable and the silica in the toothpaste could be either ES or EP.*

Answer: Thanks for your suggestion. We agree that ES could not be excluded with TP samples. In order to obtain more accurate results, we first added three- and four-class LDA-based classifiers in this revision as you suggested (see the response to the Question 5).

Then, we have used different classifiers to analyze the sources of SiO₂ NPs in real samples. Considering that ES could not be well differentiated from EP, the source of SiO₂ NPs in TP samples was estimated by using the four-class LDA model in which EP and ES were treated as one source. Result showed that the sources of SiO₂ NPs in TP samples were highly probable to originate from EP or ES (probability 90.0–96.0% for EP+ES in four-class LDA). Please see page 11 line 225–page 12 line 228.

3. **Question:** *Line 229-230: ‘This makes it essentially different from currently used isotopic labelling methods and thus suitable for source tracing in complex systems (e.g., natural environment, biological, and industrial systems).’ & Line 247 – 249: ‘The distinguishable isotopic fingerprints provide a practical approach for label-free detection of engineered NPs against high background signals in the natural environment.’ – I do not think this is a valid claim based on the data presented in this paper. There is a strong argument to be made that, when a sample is 100% natural or engineered, the isotope signature may be used to distinguish between the two. However considering the variability in isotope composition of both engineered and natural sources and the magnitude of the differences it seems highly unlikely that this would be possible. It is also likely that the relative proportion of engineered to total silica in a natural environmental sample will be too small to result in difference that is resolvable from the natural variability. These claims either need to be toned down or justified by theoretical mixing calculations or analysis of environmental samples.*

Answer: Thanks for your suggestion. We agree that it may be premature to claim that the distinguishable isotopic fingerprints permit the label-free detection of engineered NPs against high background signals in the natural environment. This statement has been removed from the revised manuscript as it actually is not within the scope of this work. Please see page 12 line 238. Furthermore, we have also carefully checked the whole manuscript to avoid any overstatements.

4. **Question:** *Table 1: - the number of correctly identified natural NPs should be 35 not 15.*

Answer: The number of correctly identified natural NPs has been corrected to 35 in Table 1.

5. **Question:** *Figure 2e – This represents the main output of the presented work and is essentially the same data as that in supplementary figure 7, with a few additional data points. In figure S7 the graph is split into three fields – two natural and one for engineered. Have the authors considered including these zones/lines in figure 2e? Using the criteria in figure 7 there is a 100% success in distinguishing between natural and engineered and only 93.3% when using the machine learning approach so is this arguably more reliable than the LDA? From figure 2e it is apparent that sol-gel and precipitated silica are indistinguishable, and hence why the LDA identifies 5 of the 7 sol-gel samples as precipitated silica and the resulting contours for sol-gel silica do not seem logical. This may be because the model is*

trying to force a difference where none exists. Have the authors performed the LDA by with different groups – NQ – ND – ES+EP+EF and/or NQ – ND – EF – ES+EP?

Answer: Thanks for your suggestion. First, we have added the old Supplementary Fig. 7 in the main paper (now as Fig. 2e and 2f) that allows an easy comparison with the LDA method.

Second, compared with the line-splitting method, the LDA-based machine learning approach has at least three advantages: i) it can provide quantitative results for the source distinguishing of SiO₂ NP samples; ii) it can give the probability for every potential source (rather than only one), so it should be more rigorous than the line-splitting method; iii) it can be easily extended to other types of NPs with more than two elements (i.e., three or more dimensional system), which will no longer be split into isolated zones by simple lines. Therefore, although the discrimination accuracy of LDA did not achieve 100%, it should be more practical and precise than the line-splitting method. These features of LDA have been introduced in the revised manuscript. Please see page 8 line 155-page 9 line 162.

Third, according to your suggestion, we have also tried the LDA with three classes (NQ / ND / ES+EP+EF) and four classes (NQ / ND / EF / ES+EP). The results are given in Fig. 2g, Supplementary Fig. 7, and Supplementary Table 3-7. These new results made the technique more comprehensive. Furthermore, we have also applied all the three LDA models in real sample analysis. Please see Supplementary Fig. 9 and Supplementary Table 11 and 12.

6. **Question:** As a side comment I would encourage the authors to consider how this plot might be made easier to read for someone who is colour-blind – for example using different shapes for data points instead of just different colours.

Answer: According to your kind suggestion, we have used different shapes in addition to different colors to differentiate data points in all plots in the manuscript and SI. This indeed makes the plots more readable.

7. **Question:** Figure 4 caption Lines 520, 521, 522 – consider removing ‘natural’ from ‘natural Si and O isotopic ...’ To remove any possible confusion as you refer to natural silica and engineered silica.

Answer: Thanks for your suggestion. We have removed “natural” from any places where it may cause confusion. Please see the captions of Fig. 2, Fig. 4, Supplementary Fig. 6, and Supplementary Fig. 8.

Finally, we thank again the editor and the reviewers for your great efforts on improving the quality of this manuscript. We are looking forward to hearing your decision soon.

Thank you very much!

Best wishes,

Yours sincerely,

Dr. Qian Liu and Dr. Guibin Jiang

REVIEWERS' COMMENTS:

Reviewer #1 (Remarks to the Author):

Overall, the authors have adequately taken into account the issues raised by the two referees during a first round of reviewing. In my opinion, the revision made has further improved the quality of the paper. Given the novelty of the approach described and its potential relevance in the field of nanomaterial analysis, I would like to recommend this paper for publication in Nature Communications, provided that the following (mostly minor) remarks are addressed.

Title. I am still not convinced that the title adequately describes the content of the paper. An analysis focusing on the determination of 2 elements would never be referred to as a multi-element method. Therefore, I would rather suggest 'Distinguishing the sources of nanoparticles based on the combined isotopic fingerprints of Si and O' or something along these lines.

Abstract. The abstract only mentions isotope fractionation during the manufacturing process as the source of differences in the isotopic composition of Si and O between NNPs and ENPs. Later on, the authors hypothesize that in the case of O, also the use of isotopically heavier reagents in the manufacturing of ENPs plays a role. This should be mentioned in the abstract as well.

Abstract – as the term 'nanoanalytics' is not interpreted in the same way by all scientists, I would recommend using 'nanomaterial analysis' instead. As an illustration, a recent book on the topic describes nanoanalytics as "a novel branch of analytical chemistry which explores applications of nanotechnologies in chemical analysis" (<https://www.degruyter.com/viewbooktoc/product/487908>).

Abstract – line 17. I suggest replacing 'challenges' by 'shortcomings'.

Line 138. Important: Is the (relative) difference in natural abundance between two isotopes affecting the extent of isotope fractionation? I don't think so. Otherwise isotope ratios of an element with three (or more) isotopes would not be located on a mass-dependent fractionation line.

Line 162. I would further soften the statement by replacing 'can give the probability' by 'can theoretically enable the source of each source of SiO₂ NPs to be revealed' or 'can in principle enable the source of each source of SiO₂ NPs to be revealed'.

Line 267. I wonder for which elements high-precision methods for measuring their isotopic composition are still lacking?

Some editorial work is required to overall polish the text, but I would like to point out a few occasions, where the actual meaning is affected:

- Line 39: please replace 'some' by 'sometimes'.
- Line 41: please replace 'isotopes' by 'isotope ratios'.
- Line 75: please replace 'the potentials of' by 'some potential for'.
- Line 100: please replace 'only microscopy measurements' by 'microscopy measurements only'.
- Line 169: please replace 'for EP and EF' by 'between EP and EF'.

ESI – line 75. The authors mention the use of recycled material in the production of SiO₂ ENPs and mention that this may bring 'uncertainties to the isotopic fingerprints'. In fact, this is a huge threat for the applicability of the method. EF SiO₂ ENPs are characterized by the lightest Si isotopic compositions. This means that left-over material is isotopically heavier. If that material would then be re-used in the manufacturing process, this would obliterate the characteristically light Si isotopic composition of the end product!

ESI – Figure S6 aims to show absence of an influence of the NP size on the isotopic characteristics requires, but is a bit of a puzzle. I suggest to plot $\delta^{30}\text{Si}$ and $\delta^{18}\text{O}$ as a function of NP size instead.

Reviewer #2 (Remarks to the Author):

The authors have done a great job at responding to the comments of the reviewers and I feel the manuscript is now stronger and more robust. I have some minor comments and suggestions I would like to make to the authors for their consideration.

I would like the authors to consider if more appropriate phrasing than 'natural multi-isotopic fingerprinting' (as used in the title) could be used. The authors use this phrasing in the title and Line 63, and throughout the manuscript. Firstly I would suggest not using 'natural' in these instances as the authors are distinguishing engineered sources from natural, the only way this would be possible is if the isotope signatures in engineered sources are non-natural? I would therefore suggest just removing 'natural' in such instances or perhaps an alternative, such as 'inherent', would be more accurate and appropriate. The use of multi-isotopic fingerprint is also used throughout but perhaps an alternative could be used that would more accurately describe the work here. It is not really possible to have a mono-isotopic fingerprint so aren't all isotope finger prints multi-isotope? Would 'dual element isotope fingerprinting' be more appropriate?

An important and novel part of this work was the machine learning, however this is not mentioned in the title and key words. Have the authors considered incorporating 'machine learning' and maybe 'engineered' into the title. In the keywords I would suggest considering including machine learning and combining 'stable isotope' and 'fingerprint' to just 'isotope fingerprint'.

Line 74 'therefore breaks through the past perception on stable isotopic tracing of NPs' – suggest that the authors clarify that they mean stable isotope tracing using inherent isotope fingerprints – stable isotope tracing has successfully been performed by many researchers using artificially enriched stable isotopes.

Response to Reviewer Comments

Response to Reviewer #1:

Overall, the authors have adequately taken into account the issues raised by the two referees during a first round of reviewing. In my opinion, the revision made has further improved the quality of the paper. Given the novelty of the approach described and its potential relevance in the field of nanomaterial analysis, I would like to recommend this paper for publication in Nature Communications, provided that the following (mostly minor) remarks are addressed.

Title. I am still not convinced that the title adequately describes the content of the paper. An analysis focusing on the determination of 2 elements would never be referred to as a multi-element method. Therefore, I would rather suggest “Distinguishing the sources of nanoparticles based on the combined isotopic fingerprints of Si and O” or something along these lines.

Response: Thanks for your suggestion. Taking into account the suggestions from both reviewers, the title is now revised to “Distinguishing the sources of silica nanoparticles by dual isotopic fingerprinting and machine learning” to describe the study more properly.

Abstract. The abstract only mentions isotope fractionation during the manufacturing process as the source of differences in the isotopic composition of Si and O between NNPs and ENPs. Later on, the authors hypothesize that in the case of O, also the use of isotopically heavier reagents in the manufacturing of ENPs plays a role. This should be mentioned in the abstract as well.

Response: Thanks for your suggestion. We have mentioned “the use of isotopically different materials” in the Abstract. Please see page 2 line 24-25.

Abstract - as the term “nanoanalytics” is not interpreted in the same way by all scientists, I would recommend using “nanomaterial analysis” instead. As an illustration, a recent book on the topic describes nanoanalytics as “a novel branch of analytical chemistry which explores applications of nanotechnologies in chemical analysis” (<https://www.degruyter.com/viewbooktoc/product/487908>).

Response: In the Abstract, the term “nanoanalytics” has been changed to “nanotechnology risk assessment” to avoid misunderstanding. Please see page 2 line 18-19.

Abstract - line 17. I suggest replacing “challenges” by “shortcomings”.

Response: In the Abstract, line 18, the “challenges” has been changed to “shortcomings”.

Line 138. Important: Is the (relative) difference in natural abundance between two isotopes

affecting the extent of isotope fractionation? I don't think so. Otherwise isotope ratios of an element with three (or more) isotopes would not be located on a mass-dependent fractionation line.

Response: In page 7 line 139-140, the statement “large difference in mass and natural abundance between ^{16}O and ^{18}O ” has been corrected to “large difference in mass between ^{16}O and ^{18}O ”.

Line 162. I would further soften the statement by replacing “can give the probability” by “can theoretically enable the source of each source of SiO_2 NPs to be revealed” or “can in principle enable the source of each source of SiO_2 NPs to be revealed”.

Response: In page 9 line 163-164, the related statement has been changed to “theoretically enable the each source of SiO_2 NPs to be revealed”.

Line 267. I wonder for which elements high-precision methods for measuring their isotopic composition are still lacking?

Response: For some elements relevant to NPs, their isotopic analysis is still challenging due to the lack of certified isotopic standard materials and strong mass interference in the MS measurement (e.g., Ti, Pd; see Ref. 46). To make the statement more accurately, it has been revised to “for some elements it is still difficult to precisely measure their stable isotopic compositions”. Please see page 13 line 266-267.

Some editorial work is required to overall polish the text, but I would like to point out a few occasions, where the actual meaning is affected:

- *Line 39: please replace “some” by “sometimes”.*
- *Line 41: please replace “isotopes” by “isotope ratios”.*
- *Line 75: please replace “the potentials of” by “some potential for”.*
- *Line 100: please replace “only microscopy measurements” by “microscopy measurements only”.*
- *Line 169: please replace “for EP and EF” by “between EP and EF”.*

Response: The corrections have been done as you suggested and we have endeavored to check the manuscript to avoid ambiguous statements in the manuscript.

ESI - line 75. The authors mention the use of recycled material in the production of SiO_2 ENPs and mention that this may bring “uncertainties to the isotopic fingerprints”. In fact, this is a huge threat for the applicability of the method. EF SiO_2 ENPs are characterized by the lightest Si isotopic compositions. This mean that left-over material is isotopically heavier. If that material would then be re-used in the manufacturing process, this would obliterate the characteristically light Si isotopic composition of the end product!

Response: Thanks for your comment. We have considered the recycle of unreacted materials in the production process of engineered NPs. First, the reaction **10** is a one-way irreversible reaction and the recycle of unreacted materials should only affect a few batches of products. Furthermore, such an effect has actually been included in our results. As shown in Fig. 2a, the $\delta^{30}\text{Si}$ values of some EF samples were indeed close to that of ND (i.e., the raw material), which, however, did not affect the validity of the method. Despite that, the point raised by this reviewer has been mentioned more clearly in the revised manuscript. Please see page S4 line 67 in the SI.

ESI - Figure S6 aims to show absence of an influence of the NP size on the isotopic characteristics requires, but is a bit of a puzzle. I suggest to plot $\delta^{30}\text{Si}$ and $\delta^{18}\text{O}$ as a function of NP size instead.

Response: Thanks for your suggestion. The Figure S6 has been revised to plot $\delta^{30}\text{Si}$ and $\delta^{18}\text{O}$ as a function of particle size as you suggested.

Response to Reviewer #2:

The authors have done a great job at responding to the comments of the reviewers and I feel the manuscript is now stronger and more robust. I have some minor comments and suggestions I would like to make to the authors for their consideration.

I would like the authors to consider if more appropriate phrasing than “natural multi-isotopic fingerprinting” (as used in the title) could be used. The authors use this phrasing in the title and Line 63, and throughout the manuscript. Firstly I would suggest not using “natural” in these instances as the authors are distinguishing engineered sources from natural, the only way this would be possible is if the isotope signatures in engineered sources are non-natural? I would therefore suggest just removing “natural” in such instances or perhaps an alternative, such as “inherent”, would be more accurate and appropriate. The use of multi-isotopic fingerprint is also used throughout but perhaps an alternative could be used that would more accurately describe the work here. It is not really possible to have a mono-isotopic fingerprint so aren't all isotope finger prints multi-isotope? Would “dual element isotope fingerprinting” be more appropriate?

Response: Thanks for your suggestion. First, to avoid confusion, the word “natural” before “isotope” has been removed or changed to “inherent” throughout the manuscript. Second, in the title, according to your and the 1# reviewer's suggestion, the “multi-isotopic” has been changed to “dual isotopic” to be more accurate.

An important and novel part of this work was the machine learning, however this is not mentioned in the title and key words. Have the authors considered incorporating “machine

learning” and maybe “engineered” into the title. In the keywords I would suggest considering including machine learning and combining “stable isotope” and “fingerprint” to just “isotope fingerprint”.

Response: Thanks for your suggestion. The “machine learning” has been added to the title and the keywords.

Line 74 “therefore breaks through the past perception on stable isotopic tracing of NPs” - suggest that the authors clarify that they mean stable isotope tracing using inherent isotope fingerprints - stable isotope tracing has successfully been performed by many researchers using artificially enriched stable isotopes.

Response: In page 5 line 75, the “stable isotopic tracing” has been clarified to “inherent stable isotopic tracing”.

Finally, we thank again you and the reviewers for your great efforts on improving the quality of this manuscript. We are looking forward to hearing your decision soon.

Thank you very much!

Best wishes,

Yours sincerely,

Dr. Qian Liu and Dr. Guibin Jiang